

**Two years of satellite-based carbon dioxide emission quantification at the world's**
**largest coal-fired power plants**
Daniel H. Cusworth[1,2], Andrew K. Thorpe[3], Charles E. Miller[3], Alana K. Ayasse[1], Ralph Jiorle[1],
Riley M. Duren[1,2,3], Ray Nassar[4], Jon-Paul Mastrogiacomo[5], and Robert R. Nelson[3]
[1]Carbon Mapper, Pasadena, CA, USA
[2]Arizona Institutes for Resilience, University of Arizona, Tucson, AZ, USA
[3]Jet Propulsion Laboratory, California Institute of Technology, Pasadena, CA, USA
[4]Environment & Climate Change Canada
[5]University of Toronto
Corresponding Author: Daniel H. Cusworth (dan@carbonmapper.org)
**Abstract**

14       Carbon dioxide ($CO_2$) emissions from combustion sources are uncertain in many places

across the globe. Satellites have the ability to detect and quantify emissions from large $CO_2$ point
sources, including coal-fired power plants. In this study, we tasked the PRecursore IperSpettrale della
Missione Applicativa (PRISMA) satellite imaging spectrometer and the Orbiting Carbon
Observatory-3 (OCO-3) instrument onboard the International Space Station at over 30 coal-fired
power plants routinely between 2021-2022. $CO_2$ plumes were detected in 50% of acquired PRISMA
scenes, which is consistent with the combined influence of viewing parameters on detection (solar
illumination, surface reflectance) and unknown factors (like daily operational status). We compare
satellite-derived emission rates to *in situ* stack emission observations and find average agreement to
within 27% for PRISMA and 30% for OCO-3, though more observations are needed to robustly
characterize the error. We highlight two examples of fusing PRISMA with OCO-2 and OCO-3
observations in South Africa and India. For India, we acquired PRISMA and OCO-3 observations on



the same day and use the high spatial resolution capability of PRISMA (30 m spatial/pixel resolution)
to partition relative contributions of two distinct emitting power plants to the net emission. Though
an encouraging start, two years of tasking these satellites did not produce sufficient observations to
estimate annual average emission rates within low (<15%) uncertainties. However, as the
constellation of $CO_2$-observing satellites is poised to significantly improve in the coming decade, this
study offers an approach to leverage multiple observation platforms to better understand large
anthropogenic emission sources.
**1 Introduction**

Anthropogenic carbon dioxide ($CO_2$) emissions are dominated by strong discrete point

sources that result from energy generation at energy supply facilities (e.g., power plants) and
industrial facilities (Crippa et al., 2019). Fossil fuel combustion is the largest contributor to warming
trends globally since the pre-industrial era (IPCC, 2021). However, there remains uncertainty in the
total magnitude of emissions from these sectors as bottom-up emission estimates rely on reported
emission factors and activity data, which may vary in granularity and quality across countries and
provinces (Hong et al., 2017; Guan et al., 2017). Accurate $CO_2$ emission quantification is important
in light of the Paris Agreement, as participating countries must develop plans and report progress to
reduce their country's greenhouse gas (GHG) emissions (UN, 2015). Leveraging atmospheric
measurements, particularly satellite remote sensing, can help reduce uncertainty in facility-level $CO_2$
emission estimates, provided that the observations are accurate and sufficiently sample the facility in
time (Hill and Nassar, 2019). Deployed systematically with robust error characterization, this system
could be an anchor towards assessing and verifying anticipated $CO_2$ emission reductions as part of
national and global GHG emission reduction plans and agreements.





Several studies have shown that atmospheric sounding satellites can accurately quantify some
point source $CO_2$ emissions from large individual coal-fired power plants. First, the Orbiting Carbon
Observatory-2 (OCO-2; Crisp et al., 2017) is a space-based instrument that observes solar
backscattered near-infrared radiance in the oxygen *A* band (758-772 nm; 0.04 nm spectral resolution),
the weak CO2 band (1594-1619 nm; 0.08 nm spectral resolution), and strong CO2 band (2042-2082
nm; 0.10 nm spectral resolution). OCO-2 views in nadir mode over land, while sun glint mode
increases the signal over water giving measurements both land and water, and target mode to target
specific validation or calibration sites. With its 10-km wide swath, $\leq 1.3 \times 2.25$ km$^2$ pixel resolution,
and better than 1.0 ppm precision for retrievals of the column-mean dry-air mole fraction of $CO_2$
($XCO_2$) (Taylor et al., 2023), OCO-2 is sensitive to single $CO_2$ point sources that emit sufficiently
close to an OCO-2 orbital track and are spatially isolated from other major $CO_2$ sources. Using
satellite observations from OCO-2, Nassar et al. (2017) detected strong $CO_2$ enhancements in the
near vicinity of seven large coal-fired power plants and employed a Gaussian plume model emission
quantification technique to estimate emission rates for these facilities. Further study expanded the set
of facilities that could be quantified by OCO-2 (Nassar et al., 2021). Other studies have leveraged
the nitrogen dioxide ($NO_2$) retrieval capability and wide swath of the TROPOspheric Monitoring
Instrument (TROPOMI; van Geffen et al., 2020) to attribute and corroborate strong $CO_2$ signals seen
in OCO-2 observations (Hakkarainen et al., 2021; Reuter et al., 2019). The Orbiting Carbon
Observatory-3 (OCO-3; Eldering et al., 2019), the flight spare of OCO-2, has been on board the
International Space Station (ISS) since May 2019. Like OCO-2, it has been shown capable of
quantifying $CO_2$ power plant emissions. Nassar et al. (2022) analyzed nine successful OCO-3
acquisitions of the Bełchatów Power Station and found the variability in satellite-based emission
estimates agreed well with the variability in independently reported hourly power generation. Guo et



al., (2023) estimated emissions at Chinese power plants using OCO-2/3 and found close agreement
with emission inventories. OCO-3 is different than OCO-2 in that it has a two-axis Pointing Mirror
Assembly (PMA) for more agile pointing, allowing it to rapidly point off-nadir and take Snapshot
Area Mapping (SAM) mode observations over the course of two minutes. These SAMs are
approximately $80 \times 80$ km$^2$ collections of measurements and are typically over sites of interest
including cities, power plants, volcanoes, and flux towers.
Another class of remote sensing imaging spectrometers – sometimes also referred to as
hyperspectral imagers – also have been shown capable of detecting and quantifying strong $CO_2$
signals from large point sources. Thorpe et al. (2017) flew the Next-Generation Airborne/Infrared
Imaging Spectrometer (AVIRIS-NG) over a coal-fired power plant in Four Corners, New Mexico,
and detected strong $CO_2$ plumes. AVIRIS-NG observes a large range of solar backscattered radiance
(380-2500 nm), but at much coarser spectral resolution (5 nm), and high spatial resolution (e.g., 3 m
when flown at 3 km altitude). The much finer spatial resolution of AVIRIS-NG allows for improved
visualization of the origin of a $CO_2$ plume, but at the expense of fine precision for a single observed
$CO_2$ column. Still, Cusworth et al. (2021) analyzed a combination of AVIRIS-NG and the identically
built Global Airborne Observatory (GAO) at over 20 power plants in the U.S., quantified emission
rates, and found close agreement with continuous emissions monitoring (CEMS) hourly emission
observations. The study also showed a few examples of $CO_2$ plumes detected and quantified with the
satellite PRISMA imaging spectrometer (400-2500 nm; 10 nm spectral resolution; 30 m spatial
resolution; Loizzo et al., 2018).
The capacity for satellites to be leveraged as useful tools for reducing uncertainty in the global
$CO_2$ anthropogenic emission sector requires synthesis and routine tasking of a critical number of
facilities. Therefore, in this study, we tasked a subset of global coal-fired power plants routinely over



the course of two years to probe detection limits, emission quantification uncertainty, and data yields.
We tasked these facilities with both OCO-3 and PRISMA. The results, though not sufficient by
themselves to reduce uncertainty relative to bottom-up inventories significantly on an annual basis,
show a path forward for data fusion and synthesis of observations from the growing constellation of
planned $CO_2$ sensing satellites.

**2 Methods**
Table 1 lists the locations of all power plants we targeted during this study between 2021-
2022 with PRISMA. OCO-3 includes a subset of these sites as well as other fossil fuel combustion
sites as part of its list of possible targets. We identified coal-fired power plants to routinely target
using a combination of bottom-up and top-down information. Bottom-up coal-fired power plant $CO_2$
emission estimates rely on activity data, that usually includes permitted capacity of a power plant
and its operational state; and emission factors, usually estimated from the composition of the coal
that is combusted. Inventories, like the Global Energy Monitor (GEM), include this data for a large
set of coal-fired power plants across the globe (GEM, 2023). From the GEM database, we gathered
the top 10 largest bottom-up coal-fired power plants globally. We then gathered a list of top-down
TROPOMI $NO_2$ combustion hotspots, as inferred by Beirle et al. (2021). We included an additional
non-overlapping seven power plants using this dataset. Because the imaging scene size of PRISMA
is $30 \times 30$ km$^2$, some adjacent smaller power plants were imaged simultaneously along with these
larger power plants. In total, outside of the U.S., we made PRISMA observations at 27 power plants.
In the U.S., we chose 10 power plants to routinely target using reported EPA CEMS information
(campd.epa.gov): five of the top 30 emitting power plants, and five progressively lower emitters,
chosen so that we could assess satellite detection capabilities.




Table 1. Power plants that were targeted specifically by PRISMA in this study.

| Power Plant Name | Country | Latitude | Longitude | Number clear-sky observations | Number plume detections | Minimum quantified $CO_2$ emission (kt $CO_2$ h⁻¹) | Mean quantified $CO_2$ emission (kt $CO_2$ h⁻¹) | Maximum quantified $CO_2$ emission (kt $CO_2$ h⁻¹) |
|---|---|---|---|---|---|---|---|---|
| Mundra-Adani | India | 22.82 | 69.55 | 12 | 7 | 0.49±0.07 | 1.09±0.19 | 1.76±0.32 |
| Korba-Balco | India | 22.40 | 82.74 | 5 | 1 | NA* | NA | NA |
| PLN Paiton Baru | Indoneisa | -7.71 | 113.57 | 4 | 2 | NA | NA | NA |
| Craig | USA | 40.46 | -107.59 | 5 | 5 | 0.56±0.11 | 0.69±0.16 | 0.8±0.22 |
| Cumberland | USA | 36.39 | -87.65 | 1 | 0 | NA | NA | NA |
| Dry Fork | USA | 44.39 | -105.46 | 6 | 3 | 0.61±0.09 | 0.65±0.13 | 0.69±0.16 |
| H L Spurlock | USA | 38.70 | -83.82 | 5 | 3 | 1.15±0.32 | 1.26±0.39 | 1.37±0.45 |
| Ulsan Hanju (1) | South Korea | 35.49 | 129.33 | 1 | 0 | NA | NA | NA |
| Hasdeo | India | 22.41 | 82.69 | 5 | 0 | NA | NA | NA |
| Hekinan | Japan | 34.83 | 136.96 | 6 | 4 | 0.72±0.47 | 3.88±1.09 | 8.35±2.14 |
| Baotou-1 | China | 40.66 | 109.66 | 5 | 2 | 0.19±0.07 | 0.27±0.07 | 0.35±0.07 |
| Kendal | South Africa | -26.09 | 28.97 | 7 | 2 | 0.85±0.13 | 0.85±0.13 | 0.85±0.13 |
| NTPC Korba | India | 22.39 | 82.68 | 6 | 1 | 1.28±0.27 | 1.28±0.27 | 1.28±0.27 |
| Kriel | South Africa | -26.25 | 29.18 | 8 | 3 | 0.74±0.15 | 0.82±0.15 | 0.95±0.16 |
| Labadie | USA | 38.56 | -90.84 | 4 | 4 | 0.73±0.18 | 0.73±0.18 | 0.73±0.18 |
| Martin Lake | USA | 32.26 | -94.57 | 8 | 8 | 1.45±0.31 | 2±0.59 | 2.6±0.98 |
| Matimba | South Africa | -23.67 | 27.61 | 11 | 8 | 0.33±0.05 | 0.72±0.16 | 1.14±0.32 |
| Matla | South Africa | -26.28 | 29.14 | 8 | 3 | 0.33±0.05 | 0.77±0.15 | 1.37±0.27 |
| Medupi | South Africa | -23.71 | 27.56 | 15 | 12 | 0.33±0.06 | 0.83±0.19 | 1.47±0.34 |
| Mundra-Tata | India | 22.82 | 69.53 | 12 | 5 | 0.38±0.09 | 0.74±0.13 | 1.32±0.21 |
| Niederaussem | Germany | 51.00 | 6.67 | 1 | 0 | NA | NA | NA |
| Oregon | USA | 41.67 | -83.44 | 5 | 1 | NA | NA | NA |
| Paiton-3 | Indonesia | -7.71 | 113.58 | 4 | 4 | 1.54±0.37 | 3.16±0.69 | 4.78±1.02 |
| Rihand | India | 24.03 | 82.79 | 8 | 5 | 0.83±0.17 | 0.99±0.26 | 1.36±0.38 |
| Sanfeng | China | 40.66 | 109.76 | 6 | 0 | NA | NA | NA |
| Sasan | India | 23.98 | 82.63 | 9 | 7 | 0.65±0.15 | 1.01±0.24 | 1.51±0.31 |
| Sooner | USA | 36.45 | -97.05 | 6 | 3 | 1.05±0.22 | 1.05±0.22 | 1.05±0.22 |
| Togtoh | China | 40.20 | 111.36 | 2 | 2 | 0.25±0.06 | 0.91±0.17 | 1.58±0.27 |





| | | | | | | | | |
|---|---|---|---|---|---|---|---|---|
| Ulsan Hanju (2) | South Korea | 35.47 | 129.38 | 1 | 0 | NA | NA | NA |
| Vindhyachal | India | 24.10 | 82.68 | 9 | 7 | 0.33±0.1 | 0.72±0.15 | 1.24±0.23 |
| Waigaoqiao | China | 31.36 | 121.60 | 6 | 1 | NA | NA | NA |
| Yeosu Hanwha | South Korea | 34.84 | 127.69 | 2 | 0 | NA | NA | NA |
| Yosu | South Korea | 34.83 | 127.67 | 2 | 0 | NA | NA | NA |
| Al Zour | Kuwait | 28.71 | 48.37 | 12 | 0 | NA | NA | NA |

*A value of "NA" indicates that no plumes were detected at this power plant or that the emission quantification
algorithm (described in Methods) failed to quantify an emission rate.

*2.2 PRISMA tasking and quantification*
PRISMA is a tasked satellite instrument, capable of collecting around 200 30 × 30 km$^2$ targets
per day and has 20° pointing capability off nadir. Authenticated users can program single task
requests through PRISMA's web portal (prisma.asi.it), which currently allows for 13 concurrent
requests at a time per user. We specified two-week observing windows for each task request, and
configured tasks to collect if the scene-averaged solar zenith angle (SZA) was less than 70° and
forecast meteorology anticipated less than 20% cloud cover. If the orbital configuration, weather,
SZA align and there are no other conflicting or higher priority task requests, PRISMA images a
target.
For each acquired PRISMA image, we performed XCO$_2$ retrievals on all pixels within a 2.5
km radius around the power plant. We retrieve XCO$_2$ using the Iterative Maximum A Posteriori –
Differential Optical Absorption Spectroscopy (IMAP-DOAS) algorithm, as implemented in
Cusworth et al. (2021). This approach estimates XCO$_2$ by decomposing an observed radiance
spectrum into high and low frequency features between 1900-2100 nm. For high-frequency features,
we simulate atmospheric transmission of CO$_2$, H$_2$O, and N$_2$O using a Beer-Lambert approximation.
For low-frequency features (e.g., surface reflectance, aerosol scattering), we use an 8-degree
polynomial. The forward model that drives IMAP-DOAS therefore has the following form:



$$F^h(\mathbf{x}) = I_0(\lambda) \exp\left(-\sum_{n=1}^{6} s_n \sum_{l=1}^{72} A_l \tau_{n,l}\right) \sum_{k=0}^{K} a_k \lambda^k \quad (1)$$

Where $F^h$ is simulated backscattered radiance at wavelength $\lambda$, $I_0$ is incident solar intensity, $A_l$ is the
airmass factor at vertical level $l \in [1,72]$, $\tau_{n,l}$ is the optical depth for each trace gas element, $s_n$ is the
scaling factor applied to the optical depth, and $a_k$ is a polynomial coefficient ($K=8$). Optical depths
are computed by querying meteorological information for pressure and temperature from the
MERRA-2 reanalysis (Gelaro et al., 2017), and using that information to select proper HITRAN
absorption cross sections for each trace gas (Kochanov et al., 2016). To compare the model from
Equation 1 against PRISMA observed radiance ($\mathbf{y}$), we compute $F^h(\mathbf{x})$ between 1900-2100 nm at
0.02 nm resolution, then convolve the output using the PRISMA full-width half maximum, and
sample at PRISMA wavelength positions. This results in vector $\mathbf{F}(\mathbf{x})$ that is comparable to $\mathbf{y}$. The
vector $\mathbf{x}$, also called the state vector, includes scale factors for $CO_2$, $H_2O$, $N_2O$, and polynomial
coefficients: $\mathbf{x} = (s_{CO2}, s_{H2O}, s_{N2O}, a_o, \dots, a_8)$.

$XCO_2$ is retrieved from PRISMA radiance using a Bayesian optimal estimation approach

(Rodgers, 2000). Here, the optimized state vector solution, or posterior, is solved through Gauss-
Newton iteration:

$$\mathbf{x}_{i+1} = \mathbf{x}_A + (\mathbf{K}_i^T \mathbf{S}_0^{-1} \mathbf{K}_i + \mathbf{S}_A^{-1})^{-1} \mathbf{K}_i^T \mathbf{S}_0^{-1}[y - \mathbf{F}(\mathbf{x}_i) + \mathbf{K}_i(\mathbf{x}_i - \mathbf{x}_A)] \quad (2)$$

Where $\mathbf{S}_O = [\varepsilon\varepsilon^T]$ is the observation error covariance matrix defined by the instrument signal to noise
ratio (SNR), $\mathbf{x}_A$ is the prior estimate of the state vector, and $\mathbf{S}_A$ is the prior error covariance matrix.
The matrix $\mathbf{K}$, or Jacobian matrix, represents the first derivative of the $\mathbf{F}(\mathbf{x})$ with respect to the state
vector:

$$\mathbf{K}_i = \left.\frac{\partial \mathbf{F}}{\partial \mathbf{x}}\right|_{\mathbf{x}=\mathbf{x}_i} \quad (3)$$



The posterior error covariance matrix can be computed explicitly to quantify retrieval precision:
$$\hat{\mathbf{S}} = \left(\mathbf{K}_i^T \mathbf{S}_O^{-1} \mathbf{K}_i + \mathbf{S}_A^{-1}\right)^{-1} \quad (4)$$

We quantified each PRISMA plume detection using the Integrated Mass Enhancement (IME)

approach (Cusworth et al., 2021). However, we updated the masking scheme for this analysis to
produce more reliable masks for each $CO_2$ plume. Figure 1 shows the plume masking procedure for
a plume detected at the Hekinan, Japan power plant on July 19, 2021. First, we apply a background
threshold to differentiate candidate plume pixels from the background (method to quantify
background threshold described in Results section). We then group enhanced $XCO_2$ pixels into
clusters of at least 20 connected pixels. These groups are then buffered with a one-pixel dilation filter
to smooth edges and any gaps that exist in a group. Finally, each cluster is considered part of the
plume if at least one of its pixels is within 500 m of an exhaust stack.



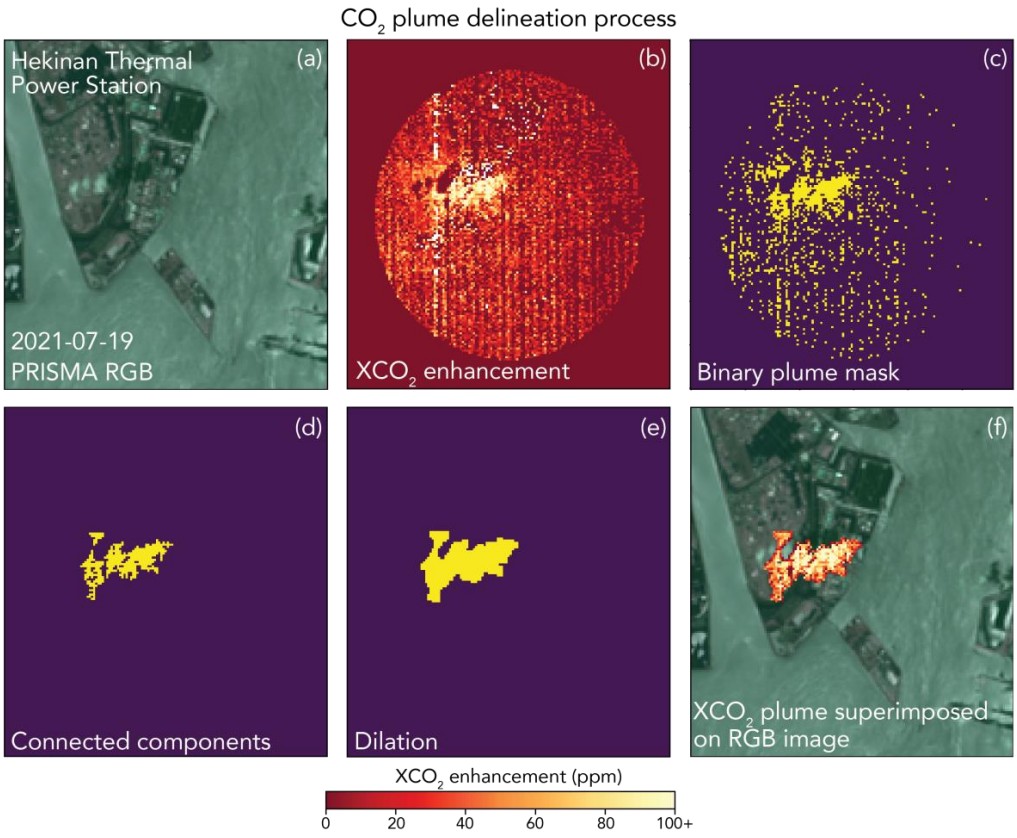

**Figure 1**. Example of the plume delineation and masking process performed on $XCO_2$ retrievals derived from PRISMA observations. Panel (a) shows the simultaneously observed RGB PRISMA imagery, panel (b) shows retrieved $XCO_2$ above the background, panels (c)-(e) show the plume masking procedure to isolate enhanced pixels and remove noise, and panel (f) shows the resulting $CO_2$ plume superimposed on the RGB imagery.

IME is calculated for a plume using the following equation:

$$IME = \sum_{i=1}^{N} \Delta\Omega_i \Lambda_i \quad (5)$$



where $\Delta\Omega_i$ is the $XCO_2$ mass enhancement in pixel $i$ relative to background (kg m$^{-2}$), $\Lambda_i$ is the pixel
area (900 m$^2$), and $N$ is the number of pixels in the plume. The $CO_2$ emission rate $Q$ is estimated from
the IME using the following relationship:
$$Q = \frac{U_{eff}}{L} \text{ IME} \quad (6)$$

where $L = \sqrt{\sum_{i=1}^{N} \Lambda_i}$ is the plume length and $U_{eff}$ is the effective wind speed, which accounts for
turbulent dissipation. We estimate $U_{eff}$ from the 10 m wind speed ($U_{10}$) using a derived empirical
relationship (Varon et al., 2018):
$$U_{eff} = 1.1 \log U_{10} + 0.6. \quad (7)$$

where $U_{eff}$ and $U_{10}$ are in units of [m s$^{-1}$]. We query the ERA5-Land reanalysis using the Open-Meteo
Application Programming Interface (open-meteo.com), which provides hourly wind information
globally at 0.1° spatial resolution (Muñoz-Sabater et al., 2021). Uncertainty due to winds is calculated
by generating an ensemble of $U_{10}$ values assuming 50% error (Cusworth et al., 2021). Uncertainty
due to the $CO_2$ background is calculated by generating many emission estimates and calculating a
standard deviation using an ensemble of background thresholds. Background thresholds are set to
vary with scene-averaged $CO_2$ retrieval precision. Total emission uncertainty is estimated by adding
in quadrature the contribution of wind and background uncertainties.

*2.3 OCO-3 tasking and quantification*

OCO-3 is also a tasked mission: it can take SAMs over any place of interest within the latitude

range of the ISS orbit (about 52° S to 52° N). In addition to the SAM locations we supplied to OCO-
3 to overlap with PRISMA targets, there are many other power plant and fossil fuel combustion
sources that make up its set of mission targets. However, unlike PRISMA, OCO-3 does not consider



cloud forecasts, snow cover, or viewing geometry when planning SAMs and thus the majority of
observations fail to produce useful maps of $XCO_2$. Additionally, aerosol- and albedo-induced XCO2
artifacts are present in many SAMs (Bell et al., 2023) and thus make the detection of plumes even
more difficult.
For all cloud-free soundings, OCO-3 XCO2 concentrations are derived using the
Atmospheric Carbon Observations from Space (ACOS; O'Dell et al., 2012; Crisp et al., 2012; O'Dell
et al., 2018) v10 optimal estimation retrieval, which employs the Levenberg-Marquardt modification
of the Gauss-Newton method. In this work, bias corrected $XCO_2$ from the OCO-3 Lite files is used
but the official data quality flag is not applied. This was done because often the quality flag removes
$XCO_2$ retrievals within the plume and makes emission estimation more difficult or impossible
(Nassar et al., 2022). For SAMs where we visually identified $CO_2$ plumes (e.g., Figure 2), emission
rates are estimated using two approaches: (1) a Gaussian plume model and (2) the IME method. For
the Gaussian plume model approach, we follow the algorithm outlined in Nassar et al. (2022):

$$V(x,y) = \frac{Q}{\sqrt{2\pi}\sigma_y(x)u} e^{-(\frac{1}{2})(\frac{y}{\sigma_y(x)})^2} \quad (8)$$


$$\sigma_y(x) = a \cdot \left(\frac{x}{x_o}\right)^{0.894} \quad (9)$$


Where $V$ represents the vertical columns within the plume ($g/m^2$), $Q$ is the $CO_2$ emission rate (g/s),
$y$ is the wind direction perpendicular to the plume (m), $u$ is the wind speed at the height of the plume
at its midline (m/s) assuming plume rise of 250 m above the stack height, $\sigma_y(x)$ is the standard
deviation of the $y$-direction, $x_o$ is a characteristic plume length (1000 m), and $a$ is a stability
parameter (Nassar et al., 2021). Following Nassar et al. (2022), wind speed and direction inputs are
estimated by taking the average of ERA-5 (Bell et al., 2020) and MERRA-2 reanalysis data. The
wind direction is optimized by rotating the plume, typically between -30° to 30° away from the mean





ERA-5/MERRA-2 direction, and calculating the correlation coefficient ($R$) of the modeled and
observed $XCO_2$. The optimized wind direction is the direction that produces the largest $R$. The
background is typically estimated by averaging OCO-3 footprints within a radius of 30 km, excluding
the plume itself and a narrow 3 km buffer zone. However, if there are visible artifacts in the $XCO_2$
background that are unrelated to the power plant plume, the background field is modified to avoid
them. For example, decreasing the radius of footprints used from 30 km to 20 km. The uncertainty
in wind speed is calculated by taking the difference of the emission estimate using two different
models (ERA-5 and MERRA2). The background concentration uncertainty is calculated by
estimating $Q$ using three different background radii of 30, 40, and 50 km. $Q$ is also calculated for a
30 km radius background but only using the left and right halves of the background, relative to the
direction of the plume. The standard deviation of both these methods is calculated and the larger of
the two is the background uncertainty. The plume rise uncertainty is calculated by estimating Q using
plume rise values of 100, 200, 250, 300, and 400 m and taking the standard deviation of those values.
Total uncertainty on the emission rate $Q$ using the Gaussian plume method is estimated by adding in
quadrature the contribution of wind speed, background concentration, and plume rise uncertainties.

To obtain another estimate of emission rate, we also apply an IME quantification approach to

the $CO_2$ plumes, which to our knowledge is the first time the IME method has been applied to OCO-
3 SAMS at coal power plants. We first interpolate the $XCO_2$ retrievals in a SAM to a uniform $2 \times 2$
$km^2$ grid to account for occasional OCO-3 footprint overlap. Similar to Varon et al. (2018), $3 \times 3$
pixel neighborhoods are sampled and the distributions are compared to the background using a
Student's $t$-test. The default confidence level for the $t$ test is 95% but this is often lowered to visually
capture most of the plume. The plume is then smoothed using a $3 \times 3$ pixel median filter and a
Gaussian filter with a standard deviation of 0.5. The $U_{eff}$ calculation is done using an equation





approximately equal to Equation 7 ($U_{eff} = 1.0 \log U_{10} + 0.55$). Other recent studies have used various
methods (Lin et al., 2023; Brunner et al., 2023), but further research is needed to determine the most
accurate way to estimate $U_{eff}$ for an OCO-3-like instrument. The wind direction is the optimized
direction determined by the Gaussian plume model. The background $XCO_2$ estimate is taken from
the Gaussian plume model methodology and the plume is typically required to be within 50 km
downwind and 8 km crosswind of the source, although these parameters are modified if the plume
curves outside of the 8 km crosswind threshold or there are $XCO_2$ artifacts that should be avoided.
The uncertainty for the IME method is estimated similarly to the Gaussian plume model
method. The uncertainty in wind speed is calculated by taking the standard deviation of the emission
estimates using wind speed from two different models (ERA-5 and MERRA2). The background
concentration uncertainty is calculated by estimating $Q$ using the different backgrounds calculated in
the Gaussian plume model method: a 20 km radius, 30 km radius, 40 km radius, left half, full circle,
and right half. The standard deviation of the three radii estimates and left half, full circle, and right
half estimates are calculated and the larger of the two is the background uncertainty. Uncertainty of
the Student's $t$-test confidence level is also estimated. The confidence level and -10% and +10% of
the confidence level are used to find $Q$. For example, if the confidence level needed to visually
capture the $XCO_2$ plume is 85%, $Q$ is calculated for 75%, 85%, and 95% and the standard deviation
of those three values represents the confidence level uncertainty. Total uncertainty on the emission
rate $Q$ using the IME method is estimated by adding in quadrature the contribution of wind speed,
background concentration, and Student's $t$-test confidence level uncertainties.
Figure 2 shows IME methodology successfully identifying an $XCO_2$ plume from an OCO-3
SAM taken over the Colstrip power plant.



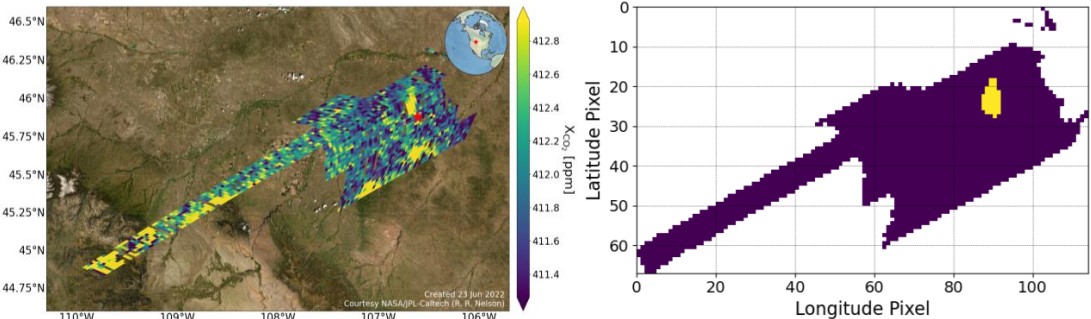

**Figure 2.** IME plume identification approach applied to an example OCO-3 SAM at the Colstrip power plant on 18 August 2021. Left panel: OCO-3 SAM bias corrected $XCO_2$. Right panel: yellow pixels indicate the final plume mask.

**3 Results**

*3.1 Global yields from two years of tasking*

Figure 3a shows a global map of power plants we targeted with PRISMA, with the marker for each power plant's location (latitude, longitude) scaled to represent the number of successful acquisitions between 2021-2022. In total, we acquired 181 PRISMA images, which corresponds to 314 unique power plant observation scenes. Of these scenes, 210 were of sufficient quality to attempt $CO_2$ retrieval and plume detection, with quality mostly determined by visual inspection for clouds and haze. Of these 210 scenes, 104 were determined to have $CO_2$ plumes (Figure 3b). Scenes were marked as containing $CO_2$ plumes through inspection of XCO2 and visible imagery: if a large cluster of pixels with elevated $XCO_2$ above the background were also in the vicinity of a power plant exhaust stack, an analyst would mark the scene as containing a $CO_2$ plume. Tasking with PRISMA resulted in an average of 6 acquisitions for each power plant (maximum 15), roughly one image acquired per



quarter. Of these acquisitions, plumes were detected on average four times per facility (maximum

12).

For OCO-3, 1363 power plant SAMs were taken during September 2019 to December 2022.

Of these, 139 $XCO_2$ plumes emanating from power plants were visually identified. However, only
14 were for power plants that were also tasked by PRISMA and have CEMS validation (nine Colstrip
cases, two Martin Lake cases, and three Craig cases). The acquisition rates are low relative to
PRISMA because OCO-3 does not account for scene favorability when planning its SAMs.  For
example, OCO-3 took 66 Colstrip SAMs from 2019-2022 yet only yielded nine high-quality $XCO_2$
plume cases.

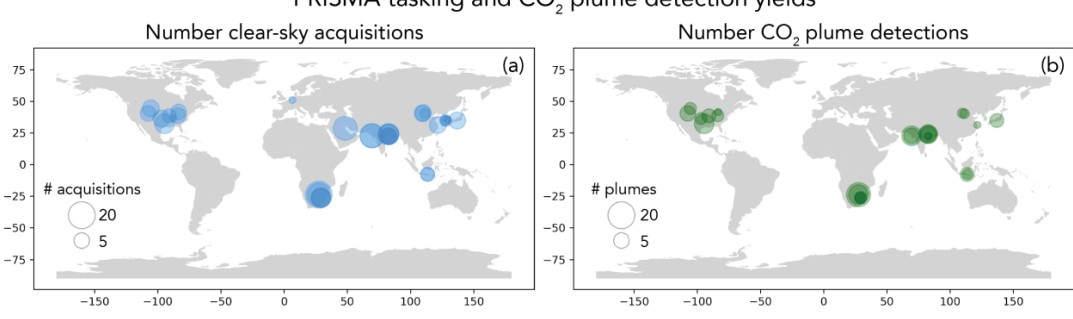


**Figure 3**. Data yields from tasking PRISMA continually between 2021-2022. Panel (a) shows the
number of clear-sky acquisitions for each power plant. Panel (b) shows the number of plumes
detected by an analyst for each of the tasked power plants.

The low observed average detection rate of plumes for PRISMA (50%) is a result of three

primary factors: (1) observing conditions at each facility including solar zenith angle (SZA) and
surface reflectance; (2) local meteorology; and (3) operational status at each power plant at the time
of acquisition. To test how well these factors predict the presence of a plume, we fit a logistic



regression classification function with a sparse (L1) penalty to our dataset (Fan et al., 2008). In this
setup, the statistical model is fit using the following predictor variables – SZA, $U_{10}$, average single-
sounding retrieval precision across the scene, annual bottom-up emission estimate for the power plant
using GEM, and average observed radiance between 1900-2100 nm within the scene normalized by
the cosine of the SZA. This last factor is a simple proxy for surface reflectance, although it does not
take into account other factors that influence radiance observations (e.g., water vapor, aerosols, other
atmospheric constituents). We split the data so that 50% was used to train the model and 50% was
reserved as a test set. The predictor variables were all standardized by their mean and standard
deviation before the model was fit. The results of classification can be summarized using two
statistics: precision (ratio of true positives to sum of true positives and false positives) and recall
(ratio of true positives to sum of true positives and false negatives). The results of fitting a logistic
regression model to the data show minor prediction performance, with precision = 0.60 and recall =
0.69 for positive plume detection. The regression coefficients are shown in Figure 4a. The coefficient
with the highest weight is normalized radiance. Figure 4b shows SZA against normalized radiance,
with red dots indicating no plume detection and blue dots representing positive plume detection.
Though no clear separation exists, there is a cluster of no plume detection at high SZA and low
normalized radiance. This is a consistent and expected relationship, as SZA and surface reflectance
are principal drivers of the quantity of light that is observed by the satellite, and therefore SNR of the
observation.



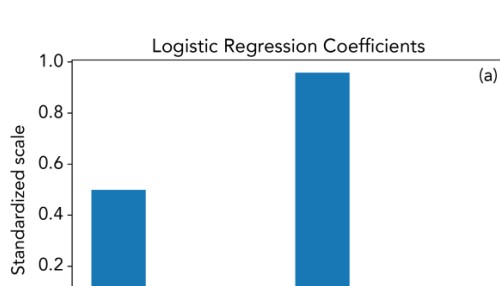
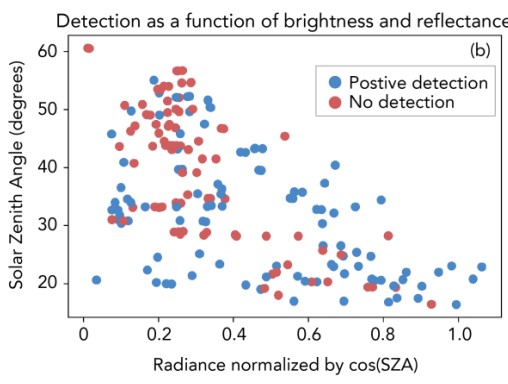


**Figure 4**. $CO_2$ plume prediction using various atmospheric, retrieval, and bottom-up information.

Panel (a) shows the results of fitting a logistic regression classification model to the set of PRISMA

acquisitions where an analyst identified the presence or lack of a plume. Panel (b) shows the top two

explanatory variables (SZA and normalized radiance) along with plume classification.

The logistic regression model performed better on the test data set than predictions made at

random, though the prediction performance was still low. Missing from the model is sub-annually

resolved information regarding operating status. For most of the power plants outside the U.S., we

do not have information on daily operations of a power plant. In many cases of non-detects, we could

simply be observing a power plant temporarily not in operation. Another possibility is that at the time

of acquisition, some power plants were operating at reduced capacity, meaning that $CO_2$ emission

rates were lower than those predicted by annual emission factors or activity data. If the true $CO_2$

emission rate was below the minimum detection limit (MDL) possible by the PRISMA satellite, then

it would show as a non-detect. However, even if the emission were near or slightly above the

PRISMA MDL, the probability of detection would still be low as slight variations in atmospheric

properties, as seen in Figure 4, would then influence the ability to detection a $CO_2$ plume.





*3.2 Validation of PRISMA and OCO-3 emission rates against CEMS*

For each power plant where a $CO_2$ plume was identified, we quantify emissions using the IME approach described by Equations 5-7. In order to estimate the $XCO_2$ mass enhancement ($\Delta\Omega$ in Equation 1), a local background must be quantified and subtracted from total $XCO_2$ retrievals across the scene. To do this, we apply a concentration threshold $\beta$ to initiate the plume masking and segmentation process (described in Methods section). Once we have a plume mask, we apply another concentration threshold $\gamma$ to the remaining $XCO_2$ pixels that exist outside of the plume. This value $\gamma$ represents the $XCO_2$ background that we use to calculate the $XCO_2$ enhancement that is used in the IME formulation of Equation 1. Thresholds $\beta$ and $\gamma$ largely influence the magnitude of the emission rate and are not known a priori. For global generalizability, we wish to estimate $\beta$ and $\gamma$ such that they do not vary across power plants, seasons, regions, etc. Therefore, we parameterize $\beta$ and $\gamma$ as percentiles under the assumption that the local contrast between enhanced $CO_2$ plume pixels and the background should be similar across PRISMA scenes.

To estimate values for $\beta$ and $\gamma$, we compare EPA CEMS data for power plants in the U.S. with estimated emission rates from PRISMA. In total, we have 12 scenes in the U.S. with CEMS information that pertain to 5 power plants. We then optimize $\beta$ and $\gamma$ such that the output of an ordinary least squares regression produces a slope near unity. Figure 5a shows the results of this optimization which produces an optimal $\beta$ percentile of 94% and a $\gamma$ percentile of 62%. The results also show decent correlation between CEMS data and PRISMA-derived emission rates ($R^2 = 0.43$). A single outlier at the Labadie power plant (imaged July 10, 2022) shows the largest discrepancy from CEMS data (69%), but the remaining plumes show average 27% relative difference from CEMS data. If we remove the one data point at Labadie, the $R^2$ improves to 0.75. Though a limited sample



size, between PRISMA and OCO-3, these scenes represent variability in solar geometries (20-40°
SZA), surface reflectance (0.09-0.90 normalized radiance), and reported emission rates (0.51 – 2.39
kt CO2 h$^{-1}$). Therefore, we use these optimal parameters and apply them to our global dataset of
PRISMA detections. These emission rates are reported in Table 1. There are some instances when
performing IME emission calculations using these thresholds and plume masking technique do not
result in emission rates (e.g., the plume masking procedure produces a mask with no pixels). In these
cases, we report a detection but not an emission quantification.

Figures 5b and 5c shows the comparison between OCO-3 and CEMS at some power plants

that overlap with PRISMA tasking (14 scenes total). OCO-3 Gaussian plume model emission rates
(Fig. 5b) have an improved correlation compared to PRISMA ($R^2 = 0.51$), although with greater bias
(average 47% relative difference from CEMS).  The OCO-3 IME estimates (Fig. 5c) have worse $R^2$
(0.32) but a better RMSE (0.45 kt $CO_2$/hr) compared to the Gaussian plume model estimates (0.84 kt
CO2/hr), with 9 of the 14 cases being within 30% of the reported CEMS emission and an average
relative difference of 30% for all 14 cases. Additionally, the least squares fit for IME is closer to the
1-to-1 line than for the Gaussian plume model.

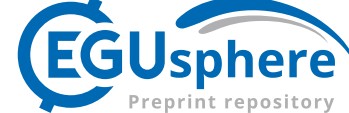

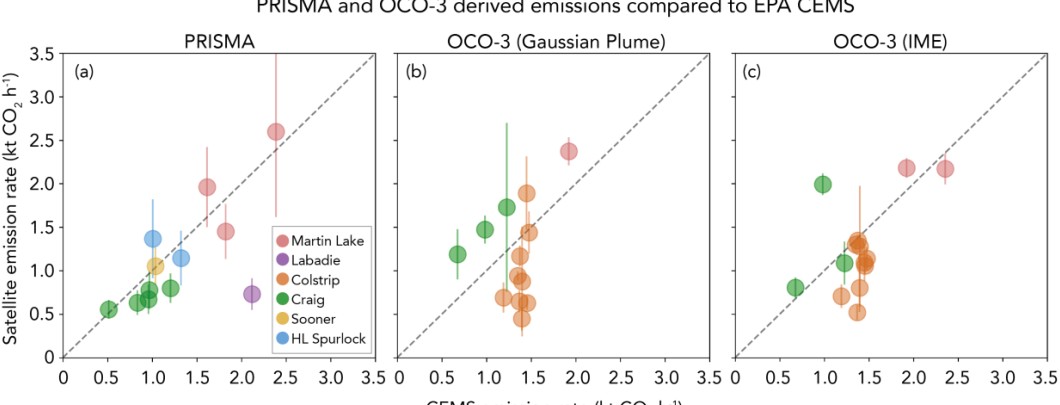


**Figure 5**. Comparison of emission rates in the U.S. between satellite-derived estimates and CEMS

information. Panel (a) shows a comparison between PRISMA derived emission rates and CEMS ($R^2$

= 0.43), panel (b) shows a comparison between OCO-3 and CEMS using the Gaussian plume model

($R^2$ = 0.51), and panel (c) shows a comparison between OCO-3 and CEMS using IME ($R^2$ = 0.32).

Unique sources of error for OCO-3 emission estimates include geolocation errors in the

XCO2 product. These errors are typically less than 1 km, but can be up to 2 km (Taylor et al., 2023).

Errors of this magnitude, about the size of an OCO-3 footprint (~2×2 km$^2$), may mean that an entire

footprint is not included when estimating emissions using the Gaussian plume method, which

assumes that the plume only extends downwind of the known source location. The Gaussian plume

model is also susceptible to wind direction errors, and requires the plume to be Gaussian in shape,

which is often not true. IME, while not suffering from wind direction or geolocation-induced errors,

assumes that the entire plume is captured in a given SAM, which is sometimes not true and results in

an underestimation of emissions. IME is also sensitive to errors in $U_{eff}$ parameterization.




*3.3 Comparison and fusion of PRISMA and OCO*


Outside the U.S., PRISMA observed the Matimba power station in South Africa 11 times and
quantified emission rates 7 times. Emissions from Matimba have previously been quantified and
validated using OCO-2 (Hakkarainen et al., 2021). This station does not report hourly emission rates,
but does report daily power generation (Eskom, 2023). Though not a direct comparison, we can use
this information to check if the emission quantification approach we describe above captures some
variability in activity at this power plant. Figure 6a shows the emission rates we quantified compared
against reported power generation. We see rough agreement in variability – the high power
generation reported between Apr to July 2021 (70000-85000 MWh) drop for subsequent dates
(47000-66000 MWh) between Sep 2021 to Sep 2022, a drop which is also seen in the PRISMA-
derived CO2 emission rate. Across all observations, we estimate an emission rate range of 0.30-1.04
kt CO2 h$^{-1}$ (average 0.66 kt $CO_2$ h$^{-1}$). This average emission rate is substantially lower than the
average 2.50 kt $CO_2$ h$^{-1}$ emission rate estimated from OCO-2 and TROPOMI between 2018-2020,
but within the range of emissions estimates directly quantified with OCO-2 (0.30-7.20 kt $CO_2$ h$^{-1}$;
Hakkarainen et al., 2021). However, this discrepancy could be result of (1) changes in activity or
variability or (2) existence of other nearby emission sources. For changes in activity, during August
2020, the Matimba reported a large range of power generation (65000-94000 MWh) and emission
estimates derived directly from OCO-2 were also highly variable (0.88-4.33 kt $CO_2$ h$^{-1}$). Given that
maximum power generation at the time of a PRISMA observation was 85000 MWh, some of the
discrepancy in maximum $CO_2$ quantification between PRISMA and OCO-2 could be due to activity.
Nearby (7 km) the Matimba Power Station is the Medupi Power Plant (Figure 6b). Figure 6c
show the Medupi $CO_2$ plume observed during the same PRISMA overpass on Apr 5, 2021. The
PRISMA derived emission rate for Medupi is 0.64 ± 0.26 kt CO2 h$^{-1}$ and for Matimba is 0.73 ± 0.30



kt $CO_2$ h$^{-1}$. Given the proximity of the two power plants, the higher derived emission rate reported
for Matimba from previous studies could actually be a result of a net emission from these two
facilities. The OCO-2 flight track is located tens of kilometers downwind from Matimba and Medupi,
making a clear delineation between potentially co-emitted distinct emission plumes near impossible.
If we sum emission rates from both Medupi and Matimba, we quantify a range of 0.89-1.73 kt $CO_2$
h$^{-1}$ (1.30 ± 0.28 kt CO2 h$^{-1}$), which is still lower, but closer to the average emissions quantified by
OCO-2.

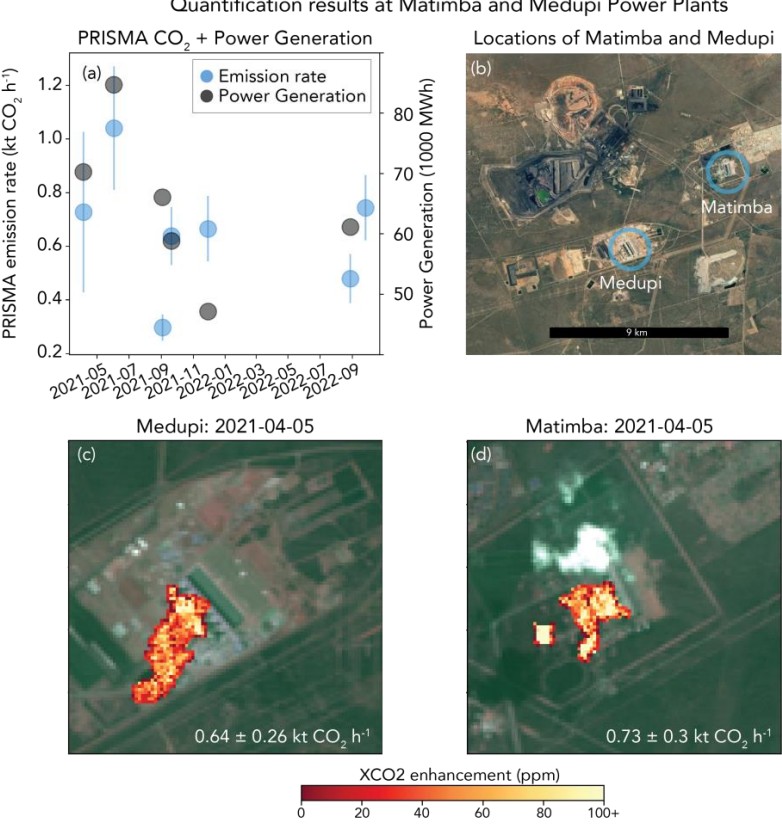


**Figure 6**. Emission rates and reported power generation at the Matimba and Medupi power plants in
South Africa. Panel (a) shows the $CO_2$ emission rates derived from PRISMA and the reported daily



power generation for the day of PRISMA overpass. Panel (b) shows the locations of the Medupi and
Matimba power plants (base imagery provided by Google Earth; © Google Earth 2023). Panels (c)
and (d) show plume imagery and emission rates for a PRISMA overpass on Apr 5, 2021.

The ability to differentiate the contribution of unique point sources to a regional total is an

application made possible by joint observing of imaging spectrometers and atmospheric sounders.
Figure 7 shows observations that were made at the Tata Mundra Ultra Mega Power Plant and the
Adani Mundra Thermal Power Project: two power plants less than 3 km apart. Both OCO-3 and
PRISMA imaged the power plants on Apr 9, 2022. Figure 7b shows the OCO-3 SAM (taken 04:41
UTC) – large $CO_2$ enhancements appear along the coastline likely associated with emission from
these power plants. PRISMA imaged the power plants less than two hours later (06:02 UTC) and
detected CO2 plumes at each facility (Figure 7b-c). The OCO-3 derived emission rate using Gaussian
plume approaches is $5.5 \pm 0.7$ kt $CO_2$ h$^{-1}$, but the emission rate derived using the IME approach is
much lower (3.0 kt $CO_2$ h$^{-1}$). For this case, the IME approach may be more appropriate as the shape
of the OCO-3 plume (Figure 7b) is more diffuse in nature and does not visibly resemble a Gaussian
structure. The PRISMA emission rate for the Adani plant is $1.07 \pm 0.17$ kt $CO_2$ h$^{-1}$ and for the Adani
plant is $0.53 \pm 0.08$ kt $CO_2$ h$^{-1}$. We can use this information to estimate that 67% of the net $CO_2$
emission came from Adani, and the remaining 33% came from the Tata plant. The combined
emission rate ($1.60 \pm 0.25$ kt $CO_2$ h$^{-1}$) is lower than the OCO-3 IME emission rate. Like the Matimba
power plant, some of this discrepancy is likely explained by uncertainty in retrievals, background,
and wind information. Continued validation of retrieved emission rates against ground standards
(e.g., CEMS) will help reduce this uncertainty. However, even with this lingering uncertainty, the




near simultaneous observations of OCO-3 and PRISMA can help us disentangle the relative
contributions from each power plant.

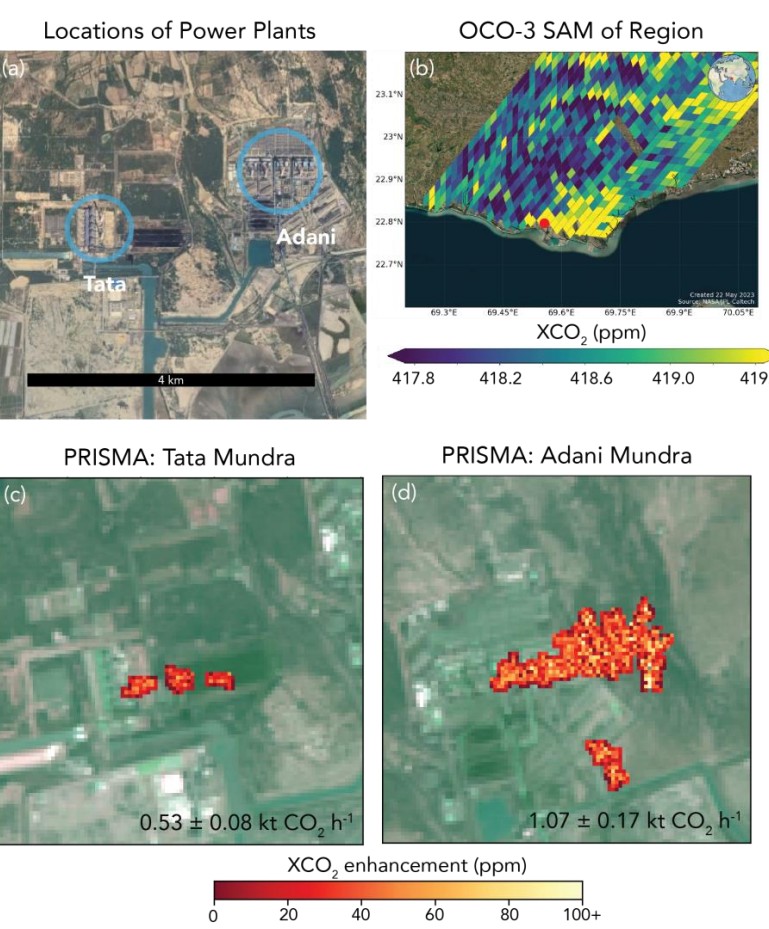


**Figure 7**. Near-simultaneous observation of two power plants in Mundra, India on Apr 9, 2022. Panel
(a) shows the locations of two power plants spaced less than 3 km apart: Tata Mundra and Adani
Mundra Power Stations (base imagery provided by Google Earth; © Google Earth 2023). Panel (b)
shows the OCO-3 SAM with a red dot showing the location of the power plants. Panel (c) and (d)





show the PRISMA acquisition (less than 2 hours after OCO-3) over the two power plants with
associated emission rates.

**Conclusion**

We tasked a global set of power plants for two years between 2021-2022 with both PRISMA

and OCO-3 to test the ability of these satellite platforms to do routine operational monitoring of $CO_2$
emissions. When PRISMA observations were of sufficient quality to perform $XCO_2$ retrievals, we
detected $CO_2$ plumes nearly half of the time. We fit a logistic regression classification using plume
detections and find that there is some relationship between SZA and surface reflectance that partially
explains plume prediction; consistent given that these factors are major drivers of SNR. The
remaining non-plume detections may be due to operational status of a power plant at the time of
observation. We compared emission rates from both PRISMA and OCO-3 to power plants in the
U.S. where we have access to hourly *in situ* CEMS emission information. We find significant
correlation between satellite and *in situ* estimates, though some significant biases may exist for some
of the observations for both PRISMA and OCO-3. Also, the quantity of CEMS observations was
limited (~10 for each instrument), so robust calibration is not yet possible. Still, early results show
that under the right conditions, satellites can provide reliable estimates of $CO_2$ emissions at discrete
point source locations. This is consistent with the close agreement between airborne imaging
spectrometer emissions and CEMS information (Cusworth et al., 2021).

Fusion of information from atmospheric sounders like OCO-3 and imaging spectrometers

like PRISMA is valuable for cross-validation and source attribution. We see this particularly for our
examples at the Matimba and Medupi power plants in South Africa and the Tata and Adani power
plants in Mundra, India. In these cases, and particularly at Mundra where near-simultaneous

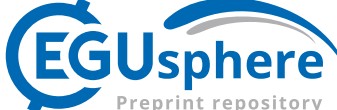

PRISMA and OCO-3 observations were taken, OCO-2/3 provides a local, but coarse resolution
emission constraint for a complex of facilities that emit large $CO_2$ quantities. PRISMA, with its 30
m pixel resolution, then can help refine relative contributions of single emitters against the net
emission flux. More work is needed to refine cross-validation between instruments, but initial tasking
shows one avenue for data from multiple observing systems to be complementary aggregated and
analyzed.

Even when combining information from both satellites, there is still too little sampling to

constrain facility emissions within low uncertainties. Cusworth et al. (2021), using arguments from
Hill and Nassar (2019), suggested that nearly 30 unbiased observations from a PRISMA-class
instrument is needed per year at each power plant to reduce annual uncertainties below 14% (i.e.,
reduce emission uncertainty from Non-Annex I countries below 1 Gt $CO_2$ per year). No power plant
in this study met this minimum sampling requirement. However, there will be a significant increase
in data volumes and observation performance of satellite remote sensing capabilities for CO2, from
both recently launched and planned imaging spectrometers including EMIT (launched 2022; Thorpe
et al., in revision); EnMAP (launched 2022; Guanter et al., 2015); Carbon Mapper/Tanager 1-2
(Planned launch 2024; Duren et al., 2021), and atmospheric sounders including CO2M (Sierk et al.,
2019). Improved observation of global power plants and emission quantification with robust error
characterization will be vital to reduce global uncertainty of anthropogenic emissions from fossil fuel
combustion sources.

**Data Availability**.
The OCO-3 XCO2 and other retrieval properties are publicly available at the NASA Goddard Earth
Science Data and Information Services Center (GES-DISC). The full suite of retrieval products in



the standard per-orbit format can be obtained at OCO Science Team et al., 2021,
https://doi.org/10.5067/D9S8ZOCHCADE. The lightweight per-day format data (Lite files), which
includes the bias corrected estimates of XCO2, can be obtained at OCO Science Team et al., 2022,
https://doi.org/10.5067/970BCC4DHH24. PRISMA data including radiance for each scene and
XCO2 retrievals is available at https://doi.org/10.5281/zenodo.8083596.

**Acknowledgments**. This work was supported by the Orbiting Carbon Observatory Science Team.
We thank the Italian Space Agency for the PRISMA satellite targets. Portions of this work were
undertaken at the Jet Propulsion Laboratory, California Institute of Technology, under contract with
NASA.

**Author Contributions**. DHC designed the study. DHC, AKA, RJ tasked and acquired PRISMA
data. DHC performed PRISMA emission quantification and validation. RRN performed OCO-3
quantification and validation. RN and JPM helped implement OCO-3 quantification algorithms. All
authors provided feedback on results and the manuscript.


**Competing interests**. The authors declare no conflicts of interest.

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
