# Peer review of "Two years of satellite-based carbon dioxide emission quantification at the world's"

_EGUsphere, 2023_

## Author Response (AR1)

We thank the reviewers for their thoughtful comments, which we have addressed below. All page and line numbers refer to those in the revised manuscript. Reviewer comments are in *italics*, our response is in plain text, and text in the revised manuscript is in blue.

**Response to Comments from Reviewer 1**

1. *The structure of the paper still needs to be improved. Sometimes it is not easy to follow. The study shows the results of PRISMA, but the main advantages of using PRISMA and IME are not very clearly emphasized.*

> We thank the reviewer for pointing this out – as use of PRISMA is the novel underpinning of this manuscript, we clarify in the text why were are using it and why it's new.

>> Line 91. From space, the PRecursore IperSpettrale della Missione Applicativa (PRISMA), launched in 2019, like AVIRIS-NG and GAO is sensitive to a large range of solar backscattered radiance (400-2500 nm), albeit at coarser spectral and spatial resolution (10 nm spectral resolution; 30 m spatial resolution; Loizzo et al., 2018). PRISMA is a tasked satellite instrument potentially capable of hundreds of $30 \times 30$ km$^2$ observations per day, with equatorial crossing time of 10:30am, and target revisit of seven days, though true revisit depends on tasking priorities of the system. Cusworth et al. (2021) showed a few examples of $CO_2$ plumes detected and quantified with the PRISMA, with quantified emissions similar in magnitude to reported CEMS emissions.

>> Line 104. To our knowledge to date, this study represents the largest satellite-based facility scale investigation of direct $CO_2$ emission quantification across a diverse set of global power plants, and the first investigation to assess the capability of PRISMA to reliably detect and quantify $CO_2$ point sources.

2. *The authors have written a nice introduction of studies using OCO2 and OCO3 but the introduction related to PRISMA is only one line. In section 2.2 the PRISMA instrument is introduced here. Detail information (spatial resolution, overpass time, etc.) of the instrument is missing. Please add more background information about PRISMA and studies using PRISMA.*

> See response to comment #1.

3. *Please check the method section. There is no section 2.1. The numbering of subsection started directly from section 2.2. Please provide a subsection to introduce some datasets used in the study, such as the GEM dataset, CEMS dataset.*

We thank the reviewer for catching this error and update the section numbering accordingly. We introduce the GEM and CEMS datasets in the beginning of section 2 so leave as is.

4. *L185, the calculation of L is assuming that the plume is square. But in reality, not many plumes are square. I assume that the L is underestimated here. This will also affect the uncertainties of emissions.*

> We clarify in the text that L, whatever its parameterization, only needs to be related to Q if one is using a Ueff parameterization.

> > Line 200. The effective wind speed relates IME and plume length parameterizations to true emission rates. This relationship can be empirically estimated through large eddy simulations using the 10-m wind speed ($U_{10}$). Here we apply the $U_{eff}$ relationship derived from Varon et al., 2018:

5. *I am wondering if there are any validation of the xco2 retrievals of PRISMA. Is there any bias of the xco2 retrievals? How much is the uncertainty? For comparison with OCO2 and OCO3, are there any systematic biases among the retrieval products?*

> $CO_2$ plume quantification relies on quantifying enhancements above a local background, meaning that scene-wide systematic retrieval biases are likely of minimum effect on emission rate quantification. What is important is that enhancements above the local background are not biased. This latter factor can not be tested through calibration sites (like TCCON), but is best tested against ground-truth emissions, like CEMS, which we describe in the text (Figure 5).

> We further clarify in the text.

> > Line 172. Across the scenes we acquired with PRISMA, using this retrieval approach, we quantify an average 3.3 ppm precision for an XCO2 column. Absolute biases in PRISMA XCO2 retrievals are less important for $CO_2$ plume detection and quantification: systematic retrieval biases are removed from a scene through the quantification and removal of a local background, as described below. To characterize bias in emission quantification, we compare emission rates derived from PRISMA to stack-level CEMS measurements (Section 3.2).

6. *When compare IME and the Gaussian Plume method for OCO-3, we clearly see that the estimates from IME clustered together (Figure 5). This means that the sensitivity of IME for OCO-3 is lower than the Gaussian Plume method. Lower resolution of OCO-3 limits the number of plume pixels. This point should be added when you described the discrepancy between IME and the Gaussian Plume method.*

We thank the reviewer for this comment, but given the small number of points where this is the case (Colstrip power plants – 2 points for IME are closer to 1-1 line / more clustered around the 1-1 line compared to GPM), we do not think we can draw this conclusion at this point.

7. *Please add DOI number in references*

We add DOIs where applicable.

8. *L31 'better understand' than what? Compare to which part? Please make it more specific.*

We clarify in the text.

Line 32. …better quantify and characterize uncertainty for large anthropogenic emission sources.

9. *L112, 'non-overlapping'. Non-overlapping with what? Please specify it here.*

We clarify in the text and change from "non-overlapping" to "unique".

10. *About table 1, please add some explain in the manuscript about why the plume detections are lower than the clear-sky observations. There are many cases that having clear-sky observation and plume detections, but there shows 'NA'. The explanation is not very clear in the text.*

We direct the reviewer to Lines 318-360 where we describe this extensively and fit a regression model to try and predict causes for the discrepancy between clear-sky observations and detection rates.

11. *L167 Please specify that the calculation of background is described in section 3.2*

We make this correction to the text.

12. *L169 'one-pixel dilation filter', do you have reference for this filter?*

We add a reference to the text.

Dougherty, E.R., 1992. An introduction to morphological image processing. In *SPIE*. Optical Engineering Press.

13. *L186-187: The Ueff calculated from Varon et al. (2018) is for methane plume observations with GHGsat instrument. Is the relationship also suitable for CO2?*

> See response to comment #4. Ueff is applied to correct for plume length (L) and IME, so is comparable for any passive tracers.

14. *L283, '210 scenes, 104 were determined.' Please explain more clearly why there are fewer determined.*

> We direct the reader to language in the text where we explain the detection process.

> > Line 299. Scenes were marked as containing $CO_2$ plumes through inspection of XCO2 and visible imagery: if a large cluster of pixels with elevated XCO2 above the background were also in the vicinity of a power plant exhaust stack, an analyst would mark the scene as containing a $CO_2$ plume.

> We also direct the reader to Lines 318-360 where we discuss extensively why we only yielded a 50% plume detection rate – and how factors like SZA, surface reflectance, etc., impact detection.

15. *L 306, it would be better to mention one or two sentence of the 'logistic regression classification function' cited from (Fan et al., 2008).*

> We clarify in the text.

> > Line 322. This algorithm fits a logit function to the plume detection outcome of each scenes (i.e., detected plume = TRUE, no detected plume = FALSE) using a set of predictor variables that are likely candidates to explain prediction results.

16. *L446-448: Please check the sentence and rewrite it. You mentioned the Adani plant two times.*

> We make the change in the text from "Adani" the second time to "Tata Mundra."

17. *Lin 448-451: The uncertainties of PRISMA look very low here: only 16%. You explain that the discrepancy is caused by the uncertainty. The main discrepancy of emission rates between PRISMA and OCO3 is that there are more emission sources mixing in the detected plume of OCO3. We see this in Figure 7b that the XCO2 enhancement around red point is among the CO2 plumes of all power plants and industrial area in Mundra. Because OCO3 has coarse spatial resolution, and this is an important reason for the difference. This should be mentioned here.*

We thank you for drawing attention to this issue. We have corrected the text. The reported uncertainties for PRISMA are truly precision metrics (i.e., variability in wind speeds, algorithms, etc). We shouldn't directly conflate that with accuracy in comparison with OCO-3. We update the text in Lines 468 and 473 to read "..by bias and uncertainty" instead of just "..uncertainty."

We also make the explicit point the reviewer suggests about comparison in emission rates between sensors.

> Line 469. Also, lower estimates of $CO_2$ emissions from PRISMA are consistent with the fact that PRISMA is only sensitive to emissions at two exhaust stacks, while the OCO-3 observation includes all $CO_2$ sources in the industrial area around Mundra

**18.** *L 451-452 Validation won't reduce the uncertainties but only to better quantify the uncertainties.*

We change the language in the text.

> Line 471. Continued validation of retrieved emission rates against ground standards (e.g., CEMS) will help better quantify uncertainty.

**19.** *Figure 7 Please add the coordinate in each sub-figure to help readers to see the scale of each map.*

The Figure has been updated.

**Response to Comments from Reviewer 2**

We thank Reviewer #2 for their thoughtful reviews. We address each of their comments below:

**1.** *lines 16-19 of abstract and other locations in the text: I know that the terminology among the users of these satellite instruments frequently includes the term "tasked" to indicate what the satellite is instructed to do. However, in a manuscript which may be read by others outside the satellite community, another term would be more understandable. Here in this sentence, I would recommend revising to "In this study the PR......Station observed over 30......2021-2022." Rewording is recommended in several places were "task" or "tasked" are used.*

We thank the reviewer for this comment. We have changed "tasked" in the text to either "observed" or "made observations" wherever it occurs, except on Line 94, 134, 215,

where we state "PRISMA is a tasked instrument…" and "OCO-3 is a tasked instrument…"

2. *lines 35-36: Give some relative numbers for the importance of power plants and other sources such as motor vehicles for CO2 emissions.*

We provide more context in the text.

Line 35. Anthropogenic carbon dioxide ($CO_2$) emissions are dominated by strong discrete point sources: power and other industrial combustion are estimated to make up 59% of global anthropogenic $CO_2$ emissions with transport, buildings, and other sources making up the remaining 20%, 9%, and 12%, respectively (Crippa et al., 2022).

3. *lines 93-94: "observing" and "observed" instead of "tasking' and "tasked"*

See response to comment #1 above.

4. *line 188: If I understand correctly, Ueff would be the wind speed at plume level. I don't understand why this wind speed would be significantly smaller than U10. Please provide some explanation.*

We expound and clarify that Ueff is a parameter that essentially accounts for biases incurred by both the length of the plume (L) and turbulent transport (Varon et al., 2018). We add clarifying language in the text.

Line 200. The effective wind speed relates IME and plume length parameterizations to true emission rates. This relationship can be empirically estimated through large eddy simulations using the 10-m wind speed ($U_{10}$). Here we apply the $U_{eff}$ relationship derived from Varon et al., 2018:

5. *line 446: the second time "Adani" appears in this sentence, it should be changed to "Tatu Mundra".*

Thank you for catching this. The name has been corrected in the text.

6. *line 465: We observed a global....*

We now use this wording in the text.

---

## Author Response (AR2)

**Response to Comments from Reviewer 1**

1. *The authors have revised the manuscript and it is much more clear to follow. However, I still have one concern. L200, authors didn't provide any explanation about the calculation of L. The re-write sentences do not make it clear for me.*

We thank the review for the comment, and we update the manuscript accordingly.

Line 200. The parameter $L$ is an operational parameter that needs to be related to the extent of the plume. Since a plume dissipates in all directions due to turbulent diffusion, an explicit scaling function (i.e., an effective wind speed $U_{eff}$) that relates $L$ and 10 m wind speed ($U_{10}$) to the true emission can be derived through large eddy simulations (Varon et al., 2018)